# Do more stress and lower family economic status increase vulnerability to suicidal ideation? Evidence of a U-shaped relationship in a large cross-sectional sample of South Korean adolescents

**Tay Jeong** * 

Department of Sociology, McGill University, Montreal, Quebec, Canada

* jeongtay@gmail.com

**Data Availability Statement:** The data can be publicly downloaded at http://www.kdca.go.kr/yhs/ after entering basic personal details. The authors

## Abstract

It is widely held in socio-behavioral studies of suicide that higher levels of stress and lower levels of economic status amplify suicidal vulnerability when confronted with a proximal stressor, reflecting the traditionally prevalent understanding in health psychology and sociology that associates adverse life circumstances with undesirable mental health outcomes. However, upon reflection, there are strong theoretical reasons to doubt that having more stress or being in a more stressful environment always increases suicidal vulnerability given the occurrence of a crisis. Using large nationally representative public survey data on South Korean adolescents, I show that the association between recent psychosocial crisis and suicidal ideation often gets stronger with more favorable levels of perceived stress and improving levels of family economic status. Overall, the increase in the probability of suicidal ideation from recent exposure to a psychosocial crisis is consistently the smallest around medium levels of stress or family economic status and larger at low or high levels. A supplementary exercise suggests that the identified moderation effects operate mainly in virtue of individual-level stress or family economic status in the relative absence of contextual influences at the school level. The findings present preliminary evidence of the stress inoculation hypothesis with regard to suicidal ideation. Research on suicidal vulnerability could benefit from increased attentiveness to the mechanisms through which being in an adverse or unfavorable life situation could protect against the suicide-inducing effects of proximal stressors.

## Introduction

Traditionally, suicide research tended to study the effect of risk factors one by one, and a wide variety of risk factors have been identified and confirmed across multiple domains. In the past couple of decades, however, there has been an increasing interest in interactions between risk factors of suicidal behavior [1]. The stress-diathesis model is an influential theoretical framework that guides research on interactions between risk factors of psychiatric outcomes and has been widely adopted in the study of suicide [2–5]. The stress-diathesis model in suicidology

do not possess the right to directly distribute the data.

**Funding:** The author received no specific funding for this work.

**Competing interests:** The author has declared that no competing interests exist.

posits that suicidal outcomes are triggered by proximal stressors, which can be a psychosocial crisis or psychiatric disorder [6, 7]. Whether being affected by such proximal stressors leads to suicide or related outcomes depends on a range of predisposing conditions that constitute a person's vulnerability or "diathesis" to suicide [8, 9]. While diathesis is sometimes conceptualized dichotomously (i.e. it was present if it jointly brought about the outcome, absent if it did not), many studies work with a continuous concept of diathesis, assuming that predisposing conditions come in grades rather than breaks (i.e. as yes-or-no conditions) [7]. As such, a background risk factor can be conceptualized as a diathesis (or vulnerability, or predisposing condition) if it amplifies the effect of a proximal stressor on a suicidal outcome measured on a probabilistic or graded scale. In a regression context, diathesis or vulnerability is often operationalized as interaction effects between an explanatory variable of interest and a proximal stressor for predicting a suicidal outcome.

Although the concept of a medical diathesis originally focused on biological traits produced by genetic expression, the concept has been expanded to include psychological and social dispositions [7]. Studies of psychological dispositions to suicidal thought or behavior have frequently targeted specific psychometric constructs that measure a person's ability to overcome or endure a crisis. Diverse measures such as problem-solving skills [10, 11], coping skills [12], enhancing attributional style [13], optimistic explanatory style [14], grit [15], emotional intelligence [16], and cognitive vulnerability [17] have been found to affect a person's vulnerability to suicidal ideation, attempt, or depression so that a better score on these cognitive or personality traits is associated with a milder association between psychosocial crisis and suicide-related outcomes. In a similar vein, research generally reported that various forms of social support that offer relief in times of episodic stress decrease diathesis to suicidal outcomes. Support from family [18], friends, schoolmates [19], and local religious organizations [20] have been found or argued to mitigate the suicide-inducing effects of recent negative life events. These psychological attitudes/dispositions and social environments are sometimes called "internal" and "external" protective factors, respectively [21].

Other research on vulnerability to suicide-related outcomes takes interest in aspects of well-being or living circumstances rather than psychological or social traits specifically intended to measure a person's psychological or emotional durability. Contemporary studies of this orientation have commonly reported that people living in a more stressful state are more prone to suicide-related symptoms following recent negative life events or other important risk factors. It has been reported that the association between exposure to school bullying and suicidal ideation is increased with higher levels of perceived stress [22]; higher life stress amplifies the effect of loneliness and irrational beliefs on suicidal ideation among inmates [23], and the association between negative life events and suicidal attempt is increased in neighborhoods with higher levels of poverty [24]. It was also found that the effect of depression–a major correlate of suicidal ideation and a proximal stressor of suicide–on suicide attempt is stronger in neighborhoods where violence is prevalent [4]. Analogous patterns have been reported with depression as the outcome: Higher perceived stress was associated with a stronger association between recent negative life event and depression [25]; the effect of negative life events on depression was stronger among people with a lower SES [26] and living in a poorer neighborhood [27].

It should be noted that stress plays an important role in the existing literature for interpreting the relationship between (socio)economic status and vulnerability to suicidal outcomes. Most existing research of this type relies on the understanding that a person already under more stress or duress is less able to stave off the suicide-inducing effects of proximal stressors as can be seen in Kuiper, Olinger, and Lyons' statement that "individuals with a high level of global stress may perceive a general inability to cope with additional negative events and may view these events as completely overwhelming" [25]. Dupéré, Leventhal, and Lacourse, in their

theoretical explanation of the positive interaction effect between neighborhood poverty and recent negative life event for predicting suicidal attempt, argued that "[y]outh who are otherwise at risk for suicide could be more likely to exhibit suicidal thoughts and to attempt suicide when they are exposed daily to a stressful environment that is less likely to provide strong emotional, social and institutional resources in the face of a crisis" [24]. The common idea is that being under a lot of stress depletes one's mental resources to cope successfully with a sudden increase in stress from a traumatic episode. Such an understanding is consistent with the traditionally prevalent view in psychology that negative life circumstances entail undesirable mental health outcomes while well-being protects against them [28, 29]. Reflecting such an understanding, Zimmerman even interpreted the stress-diathesis model in suicidology as a framework that holds that "a risk factor (e.g. depression) has a greater likelihood of causing suicide under a condition of high stress" [4].

It is hard to deny in terms of theory as well as empirical data that being in a stressful or undesirable life circumstance *may* work as a diathesis to suicidal thought or behavior. However, there are strong theoretical reasons to think that this may not always be the case. The prevalent understanding that more stress or adversity leads to higher suicidal vulnerability pays little attention to the former's cumulative effect on the latter. In particular, the large accumulation of evidence for the *stress inoculation hypothesis* in recent one to two decades provides a strong theoretical reason to rethink the monotonic relationship described above.

It is now well-established that repeated exposure to stress and adversity not only leads to mental attrition and but possibly to resilience [29–35]. Such a "steeling" effect has been found for a wide range of adverse mental health outcomes among young adults or adolescents. For example, higher levels of work stress during adolescence mitigated the deleterious effect of work stress on self-esteem, self-efficacy, and depressed mood among young adults [36]. Notably, it has been repeatedly (but not always) found that psychological resilience draws a U-curve by childhood adversity: A nurturing and stable rearing environment as well as a deprived and stressful one leads to low resilience, while moderate exposure to adversity results in high resilience due to steeling effects. Such a quadratic pattern in stress inoculation among adolescents or young adults is reported for global distress, functional impairment, life satisfaction, post-traumatic stress symptoms [37], and depressive symptoms [38]. These findings are consistent with repeated findings from laboratory studies of dampened physiological reactivity among adolescents with moderate childhood adversity and heightened reactivity among those with low or high childhood adversity [39–42]. Research on adolescent/young adult life adversity and physiological stress reactivity have also reported a U-shaped pattern by different levels of family socioeconomic status [43], suggesting that ecological hardships during childhood contribute to or at least are correlated with psychological steeling. The effect of psychological steeling through life adversity has not yet been systematically explored for suicidality, but its known applicability to a wide range of adverse mental health outcomes demands that it be seriously considered in the study of suicidal vulnerability.

Stress inoculation theory in developmental and health psychology provides the most developed and proximate support for the possibility of a non-conventional relationship between psychological or economic adversity and suicidal vulnerability in the face of a crisis, but a related insight is also present in General Strain Theory from criminology [44]. General Strain Theory identifies the gap between expectation and reality as an important source of strain, which causes a range of negative emotions such as anger, depression, disappointment, and fear, ultimately precipitating delinquent behavior [44]. The thesis that a gap between expected and actual outcomes may cause delinquent behavior has been supported in empirical studies [45].

In general, better life circumstances tend to be associated with higher expectations of the positive states or the absence of negative states that one is entitled to in life. For example, people who

have experienced a lot of stress and adversity will be less likely to expect a life with few negative events, and socioeconomic status among adolescents is positively associated with their expectation of general life success [46]. It can therefore be conjectured that having had a life with fewer/milder adversity and accordingly having better expectations sometimes increases strain when confronted with a major crisis that at least temporarily reduces reality to a very adverse state. This is especially so considering that the negative emotions caused by strain such as depression and anger not only explain deviance and crime but are also known to affect certain psychopathological outcomes including suicide. These insights are already visible in (an oft-neglected part of) Durkheim's writing on anomic suicide, which, contrary to most recent research, stated that "poverty protects against suicide because it is a restraint in itself," and the rich are more prone to suicide than the poor in an economic disaster because "something like a declassification occurs which suddenly casts certain individuals into a lower state than their previous one" [47].

In sum, both stress inoculation theory and General Strain Theory provide potential reasons for which people in adverse life circumstances may at least in some cases be more mentally resilient in the face of severe episodic stress. The former posits that people living a (moderately) rough life may have had more opportunities to grow resilience through repeated exposure; the latter posits that they are likely to have lower life expectations and would therefore be less likely to be emotionally overwhelmed even when life suddenly falls to a very adverse state. It is a largely unexplored question how much these two separate strains of theory from psychology and criminology can be reduced to the same principle. At present, the stress inoculation hypothesis arguably provides a more developed challenge to the prevalent understanding on suicidal vulnerability and stress/adversity since it is backed by a substantial volume of empirical research on various psychiatric outcomes including depression, contrary to General Strain Theory that has rarely seen any empirical application beyond explaining crime and deviance.

The above review of theory and previous research suggests that the relationship between stress and vulnerability to suicidal outcomes in the face of a crisis is likely governed by multiple mechanisms, some of which may exert opposite effects. Since stress inoculation theory has never been seriously explored for suicidal outcomes, and as the U-curve model is not always identified even in stress inoculation research, it is hard to posit any particular relationship (e.g. increasing, decreasing, U-shape, И-shape [48]) with confidence at a purely theoretical level. Still, it would not be surprising to find systematic deviations from the conventional understanding that adolescents with higher levels of stress or living in a worse ecological environment would in general be more vulnerable to suicidal outcomes in the face of proximal stressors. Based on a very large nationally representative sample of South Korean adolescents, this article explores how people with different levels of perceived stress and family economic status vary in their association between recent psychosocial crisis and suicidal ideation with special attention to identifying non-monotonic interaction effects—something that previous studies of the same kind did not take into consideration. The analysis is intended to present a theory-backed empirical challenge to the prevalent but insufficiently substantiated understanding that those living in more stressful or underprivileged conditions are generally more vulnerable to suicidal outcomes in the face of a psychosocial crisis.

## Materials and methods

### Data

The statistical analysis is based on the Korea Youth Risk Behavior Survey (KYRBS), a nationally representative annual cross-sectional survey of middle and high school students in South Korea published by Korea Center for Disease Control and Prevention. The sampling has a stratified cluster design, in which the nation is divided into 110 strata, and each stratum is

allocated a sampling quota based on the number of secondary schools and classes. This way, a total of 800 schools are sampled each year from a little more than 5600 schools in the population. Within each sampled school, one class is sampled from each of the three grades, and the entire class gets to fill out an online survey in the school computer room. The average response rate in each class was around 95%. For increased power, I combine the data for 2017–2019, which produces a dataset with 179619 students nested in 2399 schools. The median number of students sampled in a school was 75, with 56 and 92 representing the .1 and .9 quantiles, respectively. Most variables in the dataset, including all those that were used for the statistical analysis of this paper, had no missing values since students were required to answer all questions in the online survey except in 2019 when students were allowed to skip a handful of highly confidential questions. Given the excellent data availability, variable choice was not affected by concerns about missing data.

## Variables

**Recent suicidal ideation.**   The dependent variable is a dichotomous measure of recent suicidal ideation based on the survey question, "In the past 12 months, have you ever seriously thought of committing suicide?" with a yes/no response.

**Perceived usual stress.**   This variable is based on a five-point Likert-scale response to the survey question, "Normally, how much stress do you feel?", with possible responses being "do not feel any," "not much," "some," "high", and "very high." Despite the word "normally" (or "usually"), responses to this question are most likely strongly affected by the adolescent's *recent* perceived stress at the time of the survey; yet, psychological well-being, of which stress is a major component, is known to have a strong diachronic continuity [49], and this variable is likely indicative of the cumulative psychological adversity experienced by each adolescent.

**Self-rated family economic status.**   This variable is based on a five-point Likert-scale response to the survey question, "How is the economic status of your family?", with possible responses being "low," "mid-low," "mid," "mid-high," and "high." This will be only a rough indicator of the objective amount of wealth possessed by the family, but it is expected that it would more closely reflect the quality of the ecological environment that the adolescent had been brought up in.

**Recent experience of severe grief or despair.**   This variable is based on the survey question, "In the past 12 months, have you ever experienced grief or despair severe enough to stop you from having a normal life for two weeks?" with a yes/no response.

**Recent experience of receiving clinical treatment after exposure to violence.**   This variable is based on the survey question, "In the past 12 months, have you ever received treatment in a clinic or hospital due to violence from friends, senior students, or adults?" In the survey, there are seven possible responses to choose from, each indicating the number of times of receiving treatment, ranging from *never* to *six times or more*. I transform this into a dummy variable with zero indicating the absence of such an experience and one the presence thereof. Despite some loss of information, several reasons justify the binary coding: The log-odds of suicidal ideation soar going from *never* to *once* but increases slowly thereafter; sample size rapidly decreases as the number of times increases; the statistical interaction with stress or family economic status—each coded as a categorical variable with four dummies—calls for a relatively simple coding of the proximal stressor. This variable can be expected to measure relatively traumatic cases of being affected by violence since receiving medical treatment indicates non-trivial physical injury and likely incurs considerable social stress for adolescents.

It is unlikely that there is any important class discrepancy in access to medical treatment in South Korean society, especially concerning the kind of treatment one would need for most

cases of violence. Universal healthcare and a series of equalizing policies implemented over the past couple of decades greatly increased healthcare accessibility for lower classes [50]. A 2014 study found a high degree of class equality in the frequency of receiving medical treatment even after adjusting for need [51].

Four additional variables are used as covariates in all regression models. *Self-rated health* is a five-point Likert-scale item ranging from "very bad" to "very good." As the effects were roughly linear, it was coded as a continuous variable and standardized. *Female* is a dummy variable with zero indicating boys and one indicating girls. The sex ratio was nearly even in the overall sample. *Grade* has six levels in total (grades 7–12, typically 13–18 years old). Although the effects of grade were not linear or even strictly monotonic, there were negligible differences in model fit or the coefficients of the focal independent variables depending on whether grade is included as a continuous variable or as a categorical variable, so the former coding was chosen for simplicity. There were minimal differences in the number of students in each grade. Finally, *residential type*–coded as a four-category nominal variable with possible values being "living with family," "living away from home in a dormitory/studio," "living in a relative's house," and "living in a nursery/orphanage" was included as a covariate. Living with family, which was set as the reference category, was by far the most common residential type applicable to around 95 percent of the respondents (dorm/studio 4.48%; relative's house 0.67%; nursery/orphanage 0.26%). Additional individual- and group-level control variables were explored such as academic performance, school type by vocational curricula, school urbanicity, short-term sleep deprivation, and physical exercise but excluded from the final presentation due to very small effect sizes and the lack of change in the coefficients of the independent variables of theoretical interest.

## Analytical strategy

A series of logit regressions are conducted with particular attention to the interaction between perceived usual stress and each type of psychosocial crisis. The results of population-averaged models are first presented, followed by analogous multilevel models. The population-averaged models identify the associations that hold across the national adolescent population and establish the main finding of this paper. The multilevel models are a supplementary exercise that checks whether the same pattern also holds within more local groupings and whether stress has any "contextual" effects on suicidal vulnerability at the school level. The same analyses are repeated with self-rated family economic status instead of stress, and the results are presented side-by-side for easy comparison at each stage of the analysis. A supplementary file contains an annotated, fully replicable R script displaying the output for all quantitative analyses included in this paper starting from raw data.

## Results

### Population-averaged models

Table 1 and Fig 1 summarize the distribution of and correlations among the key variables. Using this data, five population-averaged models were fitted with the svyglm() command in the "survey" package for R [52]. Model (1) contains all independent variables with no interaction term; model (2) adds interaction terms between perceived usual stress and experience of severe despair in the past 12 months; model (3) repeats the previous one but uses recent experience of violence as the type of crisis; models (4) and (5) repeat the previous two models using family economic status in lieu of stress. Stress and economic status were entered into the regressions as categorical variables with the middle category ("some" stress and "mid" economic status) as the reference. This coding choice results from the theoretical postulation that

**Table 1. Polychoric correlation among key variables.**

| Variables | (1) | (2) | (3) | (4) | (5) |
|---|---|---|---|---|---|
| (1). (DV) Suicidal Ideation | 1 | | | | |
| (2). Recent Despair/Grief | 0.698 | 1 | | | |
| (3). Recent Violence | 0.352 | 0.286 | 1 | | |
| (4). Perceived Stress | 0.538 | 0.532 | 0.107 | 1 | |
| (5). Self-rated Family Economic Status | -0.124 | -0.099 | -0.004 | -0.142 | 1 |

it is currently impossible to assume with confidence any particular shape of the interaction effects–I am leaving it as an open question that is to be determined by the large data (because of the very large sample size, there was an ample number of observations in each cross-section. Even the smallest cross-section—recent experience of violence and no stress—had 333 observations). The equation for, for example, model (3), which involves an interaction between violence in the past 12 months and usual stress, can be written as:

$$
\begin{aligned}
\text{logit}(P(Y = 1)) = {} & \beta_0 + \beta_1 VIOLENCE + \beta_2 STRESSveryhigh + \beta_3 STRESShigh \\
& + \beta_4 STRESSlow + \beta_5 STRESSnone + \beta_6 VIOLENCE * STRESSveryhigh + \beta_7 VIOLENCE \\
& * STRESShigh + \beta_8 VIOLENCE * STRESSlow + \beta_9 VIOLENCE * STRESSnone \\
& + \text{other covariates}
\end{aligned}
\tag{1}
$$

The right-hand side of Eq 1 can be rearranged to highlight the "slope" of violence more clearly:

$$
\begin{aligned}
\text{logit}(P(Y = 1)) \\
= {} & (\beta_1 + \beta_6 STRESSveryhigh + \beta_7 STRESShigh + \beta_8 STRESSlow + \beta_9 STRESSnone) \\
& * VIOLENCE + \beta_2 STRESSveryhigh + \beta_3 STRESShigh + \beta_4 STRESSlow \\
& + \beta_5 STRESSnone + \text{other covariates}
\end{aligned}
\tag{2}
$$

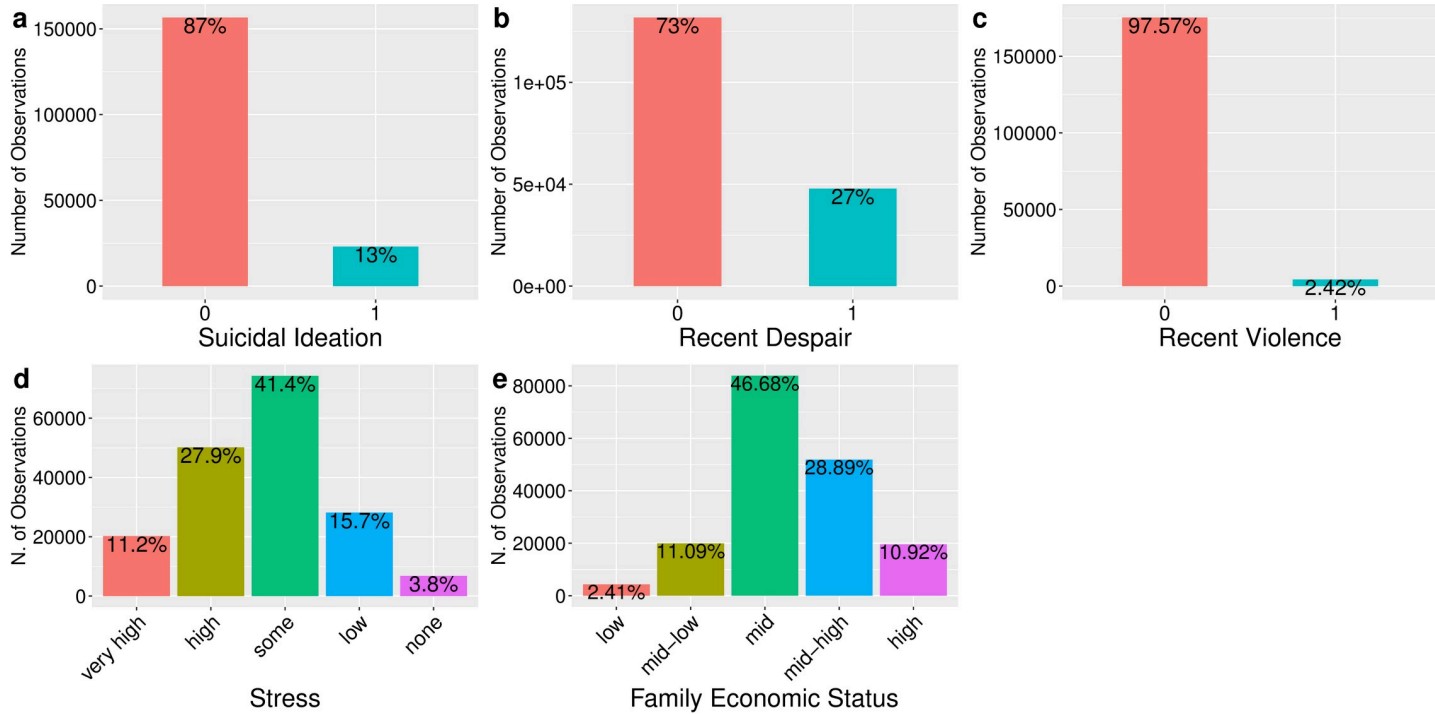

**Fig 1. Distribution of key variables.**

The regression results are presented in Table 2. The model without any interactions (model 1) had the worst fit but provides a simple model that allows easy comparison with previous research. The results of model 1 (main effects only) reveal that experience of severe grief or despair and receiving medical treatment due to violence in the past 12 months, *qua* proximal stressors, strongly increase the odds of suicidal ideation as is consistent with theoretical expectations as well as previous research [4, 53]. In addition, there is a nearly monotonically positive relationship between stress and suicidal ideation, confirming the common understanding that higher stress is a risk factor for suicidal thought [5, 25]. Likewise, lower perceived economic status was generally associated with higher odds of suicidal ideation [5], although there were negligible differences among the top three categories ("mid," "mid-high," and "high"). This is consistent with previous research that identified poverty as a risk factor for suicidal outcomes [24, 54, 55]. Overall, the main-effects-only model presents no surprise and repeats conventional knowledge.

When interaction effects are added to the base model, however, the results start to appear more surprising. Models 2–3 show that, when either type of psychosocial crisis occurs, having a higher level of perceived stress does not necessarily lower the odds of suicidal ideation, and models 4–5 show similar results for self-rated economic status (The regression coefficients of the interaction terms, analogous to $\beta_6$-$\beta_9$ in Eqs 1 and 2, are visualized in S1 Fig). Fig 2 visualizes the predicted probability of suicidal ideation for each category of stress or economic status, in the presence and absence of crisis exposure in the past 12 months, with all non-interacted covariates set to zero or the reference category. For model 2, for example, this amounts to a hypothetical student with no experience of receiving medical treatment due to violence in the past 12 months, "mid" perceived family economic status, average perceived health, male, living with family, and in 9th or 10th grade (15–16 years of age). In all four models, what looks like a slightly bent downward-sloping line in the absence of severe despair in the past 12 months is transformed into a clear quadratic shape in the presence thereof. It is to be noted that, in all four models, the absolute probability of suicidal ideation is still the highest among those with the highest stress and lowest economic status.

Figs 3 and 4 directly visualize additive and multiplicative interaction effects in a probability scale using the results shown in Fig 2. Insofar as suicidal vulnerability can be operationalized as interaction effects, Figs 3 and 4 represent two alternative conceptions of suicidal vulnerability. Fig 3 repeats Fig 2 but with relative rather than absolute risk. It shows the ratio of the dot below to the dot above in Fig 2 and represents multiplicative interaction effects in a probability scale. Relative risk increases strongly and nearly monotonically as one goes from the highest to the lowest level of stress, indicating that exposure to either type of crisis, compared to non-exposure, multiplies the probability of suicidal ideation by a larger number at lower levels of usual stress. A similar pattern is seen for family economic status, although there is an abrupt reversal in the worst (lowest) category. Note that the pattern in relative risk is similar to the pattern in the regression coefficients of the interaction terms presented in Table 1 and S1 Fig.

Fig 4 plots the *difference* in the predicted probabilities shown in Fig 2, i.e. the distance between the dot above and the dot below. This represents additive interaction effects in a probability scale. The difference tends to be the smallest at or near the middle category and gets wider towards the extremes, indicating that exposure rather than non-exposure to a crisis increases the probability of suicidal ideation by the smallest margin at or near the middle category and by larger margins at the lowest and highest categories.

Several diagnostics were conducted to check the robustness of the findings. First, two additional models that interact each type of crisis with *both* stress and economic status were run and plotted (plot shown in S1 Code). The patterns presented so far remained largely unchanged. Second, the above-visualized interaction effects were checked by dropping control

**Table 2. Population-averaged models, output.**

| | *Dependent variable*: logit(P(Suicidal Ideation = 1)) | | | | |
|---|---|---|---|---|---|
| | **(1)** | **(2)** | **(3)** | **(4)** | **(5)** |
| Recent despair/grief | 1.90*** | 2.20*** | 1.88*** | 1.87*** | 1.90*** |
| | (1.86, 1.93) | (2.14, 2.27) | (1.85, 1.92) | (1.82, 1.92) | (1.86, 1.93) |
| Recent violence | 1.14*** | 1.05*** | 1.41*** | 1.13*** | 1.03*** |
| | (1.05, 1.22) | (0.97, 1.13) | (1.27, 1.56) | (1.05, 1.22) | (0.89, 1.17) |
| STRESS "none" | -0.15* | -0.80*** | -0.46*** | -0.14* | -0.16* |
| | (-0.28, -0.03) | (-1.04, -0.56) | (-0.63, -0.28) | (-0.27, -0.01) | (-0.29, -0.04) |
| STRESS "low' | -0.61*** | -0.92*** | -0.72*** | -0.60*** | -0.61*** |
| | (-0.70, -0.52) | (-1.05, -0.79) | (-0.82, -0.62) | (-0.69, -0.52) | (-0.70, -0.52) |
| STRESS "high" | 0.87*** | 1.17*** | 0.90*** | 0.87*** | 0.87*** |
| | (0.83, 0.91) | (1.11, 1.24) | (0.86, 0.94) | (0.83, 0.91) | (0.83, 0.91) |
| STRESS "very high" | 1.58*** | 1.94*** | 1.63*** | 1.58*** | 1.58*** |
| | (1.53, 1.63) | (1.86, 2.02) | (1.58, 1.68) | (1.53, 1.63) | (1.53, 1.63) |
| Econ Status "low" | 0.47*** | 0.45*** | 0.46*** | 0.41*** | 0.45*** |
| | (0.38, 0.56) | (0.36, 0.54) | (0.37, 0.55) | (0.26, 0.57) | (0.35, 0.54) |
| Econ Status "mid-low" | 0.31*** (0.26, 0.36) | 0.30*** (0.25, 0.35) | 0.31*** (0.26, 0.35) | 0.40*** (0.32, 0.48) | 0.31*** (0.26, 0.36) |
| Econ Status "mid-high" | 0.01 | 0.01 | 0.01 | -0.06+ | 0.01 |
| | (-0.03, 0.05) | (-0.03, 0.05) | (-0.03, 0.05) | (-0.12, 0.01) | (-0.03, 0.04) |
| Econ Status "high" | 0.07* | 0.07* | 0.07* | -0.06 | 0.04 |
| | (0.01, 0.13) | (0.01, 0.12) | (0.01, 0.13) | (-0.16, 0.04) | (-0.02, 0.10) |
| Perceived health | -0.24*** | -0.24*** | -0.24*** | -0.24*** | -0.24*** |
| | (-0.26, -0.22) | (-0.25, -0.22) | (-0.25, -0.22) | (-0.26, -0.22) | (-0.26, -0.22) |
| School grade | -0.11*** | -0.11*** | -0.11*** | -0.11*** | -0.11*** |
| | (-0.12, -0.10) | (-0.12, -0.10) | (-0.12, -0.10) | (-0.12, -0.10) | (-0.12, -0.10) |
| Female | 0.22*** | 0.22*** | 0.22*** | 0.22*** | 0.22*** |
| | (0.18, 0.26) | (0.18, 0.25) | (0.18, 0.25) | (0.18, 0.26) | (0.18, 0.25) |
| Living in relative's house | 0.48*** | 0.43*** | 0.39*** | 0.48*** | 0.50*** |
| | (0.31, 0.65) | (0.26, 0.60) | (0.23, 0.56) | (0.31, 0.65) | (0.33, 0.67) |
| Living in a dorm/studio | 0.14*** | 0.12** | 0.12** | 0.14*** | 0.14*** |
| | (0.06, 0.22) | (0.04, 0.20) | (0.04, 0.20) | (0.06, 0.22) | (0.06, 0.22) |
| Living in a nursury/orphange | 0.26+ | 0.20 | 0.24+ | 0.25+ | 0.23+ |
| | (-0.002, 0.53) | (-0.06, 0.46) | (-0.01, 0.49) | (-0.01, 0.52) | (-0.04, 0.50) |
| Despair x stress "none" | | 1.45*** (1.12, 1.78) | | | |
| Despair x stress "low" | | 0.83*** | | | |
| | | (0.65, 1.01) | | | |
| Despair x stress "high" | | -0.52*** | | | |
| | | (-0.60, -0.43) | | | |
| Despair x stress "very high" | | -0.56*** | | | |
| | | (-0.65, -0.46) | | | |
| Violence x stress "none" | | | 0.88*** | | |
| | | | (0.53, 1.22) | | |
| Violence x stress "low" | | | 0.91*** | | |
| | | | (0.62, 1.20) | | |
| Violence x stress "high" | | | -0.61*** | | |
| | | | (-0.82, -0.41) | | |

(*Continued*)

**Table 2.** (Continued)

| | Dependent variable: logit(P(Suicidal Ideation = 1)) | | | | |
|---|---|---|---|---|---|
| | (1) | (2) | (3) | (4) | (5) |
| Violence x stress "very high" | | | -0.70*** | | |
| | | | (-0.90, -0.51) | | |
| Despair x Econ Status "low" | | | | 0.08 | |
| | | | | (-0.11, 0.26) | |
| Despair x Econ Status "mid-low" | | | | -0.14** | |
| | | | | (-0.24, -0.04) | |
| Despair x Econ Status "mid-high" | | | | 0.10* | |
| | | | | (0.01, 0.18) | |
| Despair x Econ Status "high" | | | | 0.20*** | |
| | | | | (0.08, 0.32) | |
| Violence x Econ Status "low" | | | | | 0.26+ |
| | | | | | (-0.04, 0.55) |
| Violence x Econ Status "mid-low" | | | | | -0.03 |
| | | | | | (-0.29, 0.23) |
| Violence x Econ Status "mid-high" | | | | | 0.06 |
| | | | | | (-0.16, 0.28) |
| Violence x Econ Status "high" | | | | | 0.37** |
| | | | | | (0.14, 0.60) |
| Constant | -3.65*** | -3.80*** | -3.65*** | -3.63*** | -3.64*** |
| | (-3.69, -3.60) | (-3.86, -3.75) | (-3.69, -3.60) | (-3.68, -3.58) | (-3.68, -3.60) |
| Observations | 179,619 | 179,619 | 179,619 | 179,619 | 179,619 |
| Akaike Inf. Crit. | 101,007.60 | 100,505.80 | 100,810.50 | 100,988.00 | 101,003.30 |

Note
+ $p<0.1$
* $p<0.05$
** $p<0.01$
*** $p<0.001$.

variables one at a time. The general shape of the interaction graphs remained largely similar throughout the variation in the controls. In addition, an examination of variance inflation factors (VIF) revealed low levels of data-based collinearity, and even most interaction terms had unproblematic levels of VIF ($<5$). The interaction terms between the two highest categories of perceived usual stress and severe despair in the past 12 months had a VIF that slightly exceeded 5 (5.6 and 5.9, respectively).

## Multilevel models

The above analysis reveals how the association between crisis and suicidal ideation varies by different levels of stress and by different levels of economic status across the national population. Does the same pattern also hold within more local groupings? When a population is divided into clusters, the variation in a raw individual-level variable is a conflation of within-cluster and between-cluster variation, and the corresponding regression coefficient will also reflect a blend of within-group and between-group relationships [56]. For example, it is conceivable that the above population-averaged analysis found adolescents with high perceived

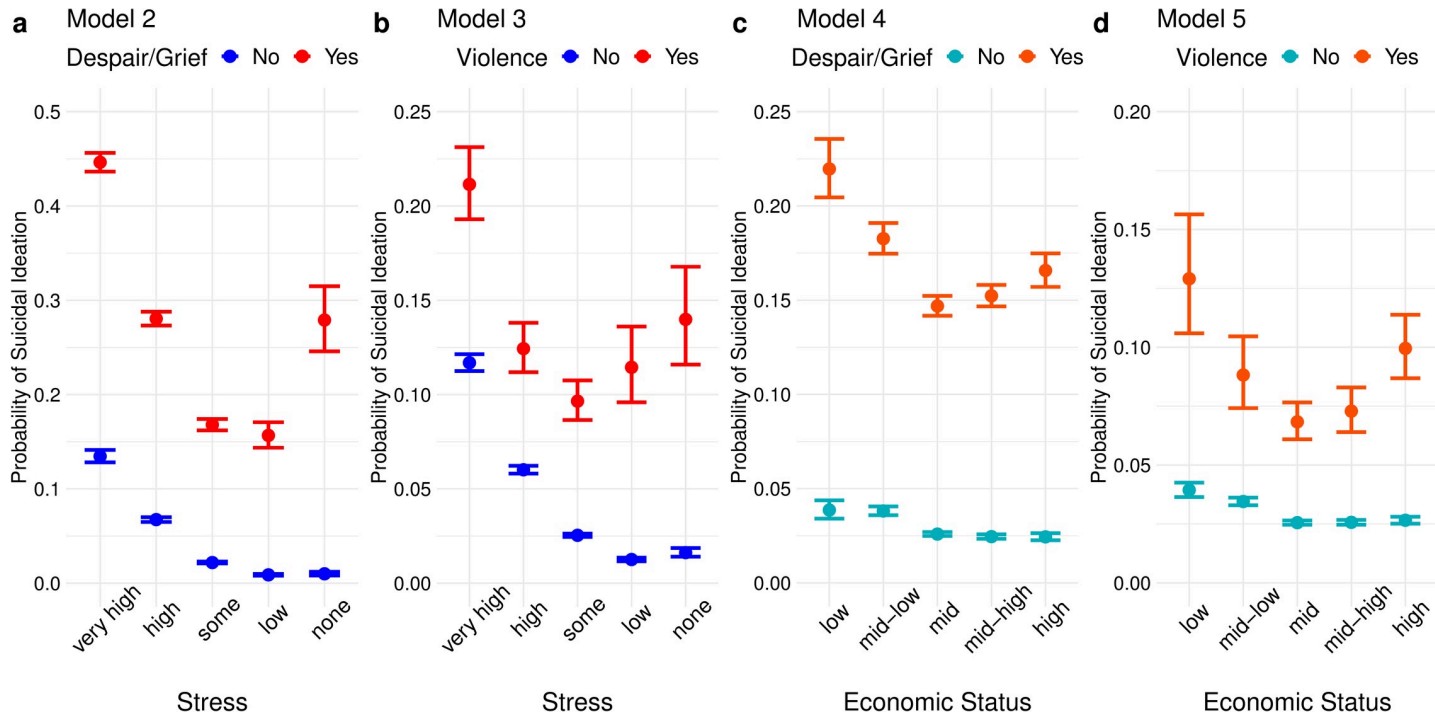

**Fig 2. Predicted probabilities for each category of stress and economic status.** All variables not involved in the interaction were set to zero or the reference category. Error bars indicate 90% confidence intervals.

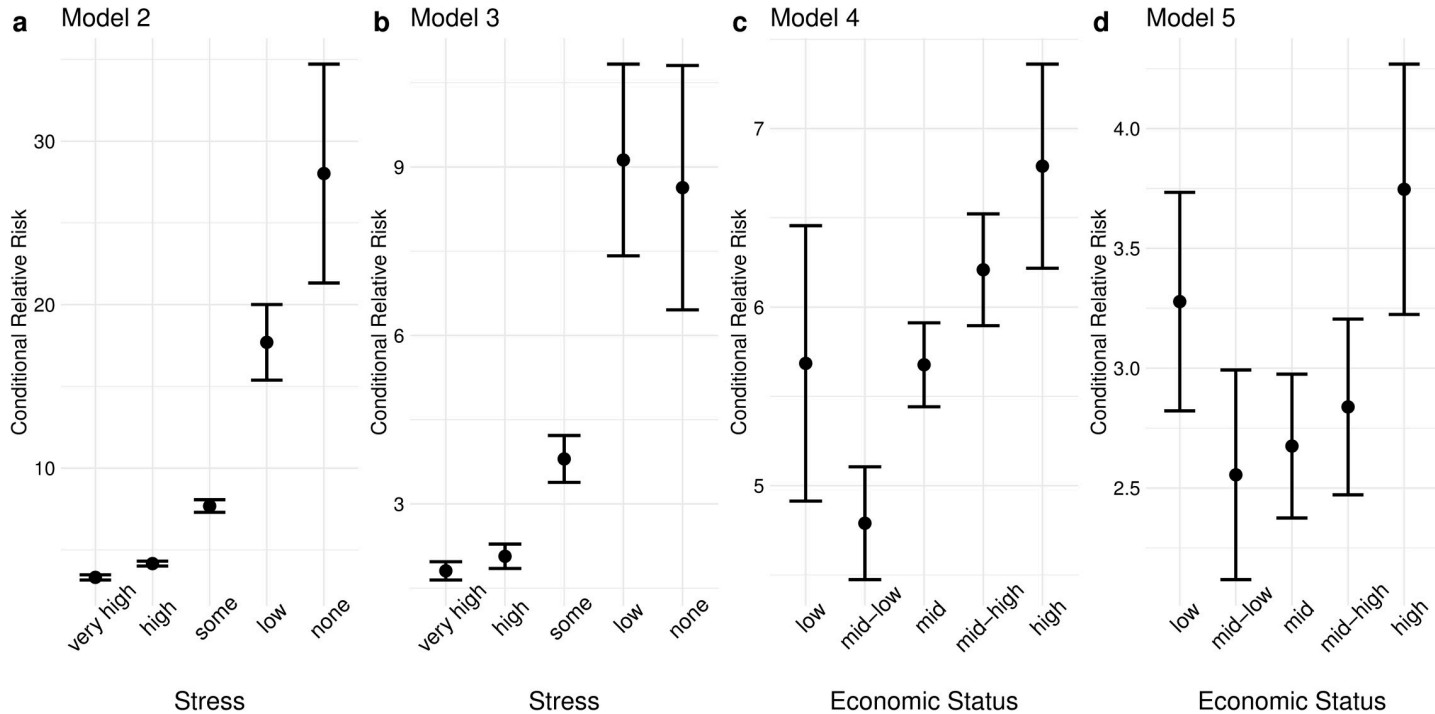

**Fig 3. Relative risk for each category of stress and economic status.** All covariates not involved in the interaction were set to zero or the reference category. Standard errors were estimated with the delta method. The error bars indicate 90% confidence intervals.

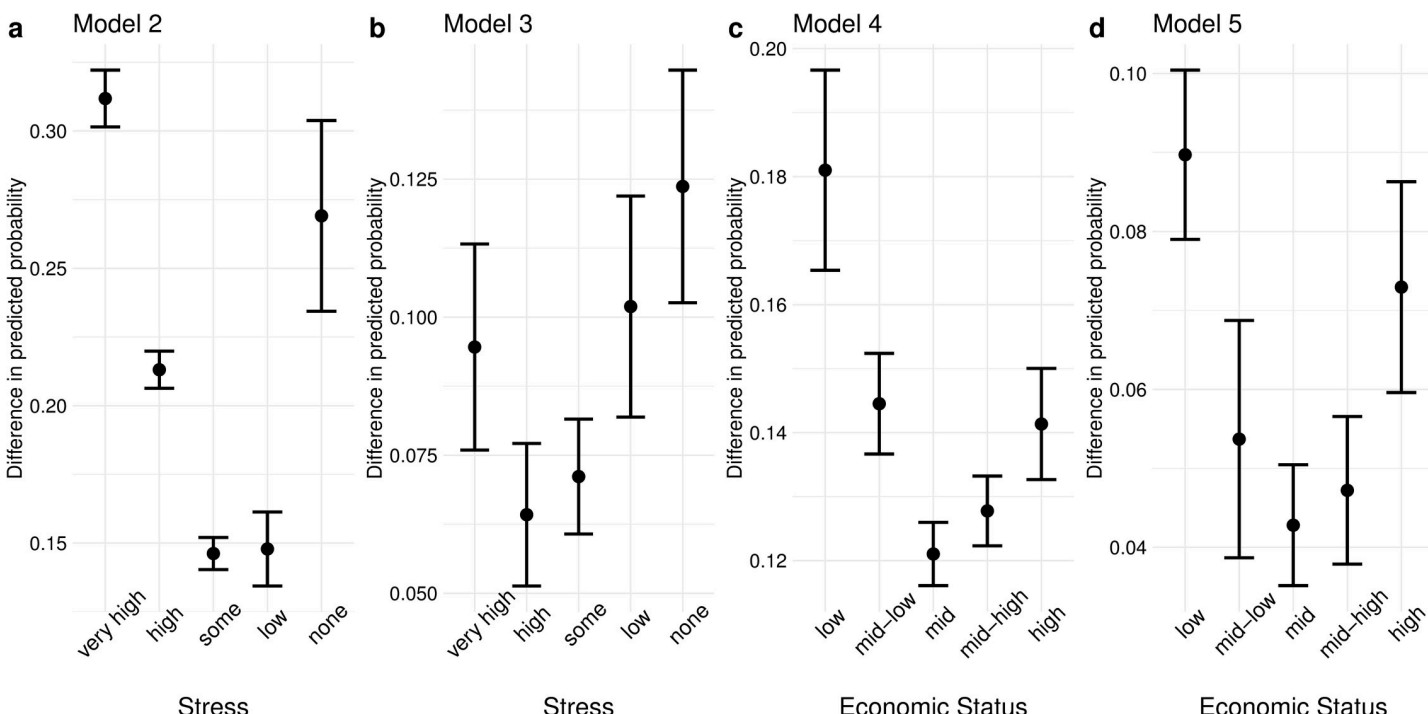

**Fig 4. Risk difference for each category of stress and economic status.** All covariates not involved in the interaction were set to zero or the reference category. Standard errors were estimated with the delta method. Error bars indicate 90% confidence intervals.

economic status to be especially vulnerable to a crisis at the national level because adolescents going to schools with a high proportion of students who think they are well-off are more vulnerable to a crisis than those going to schools where lots of students feel poor, without there being the same kind of association within each school or neighborhood.

To examine how much the pattern observed in the national population is repeated within schools, I rerun models 2–5 as multilevel models. In each, the binary crisis variable is cluster-mean centered to reflect within-cluster variation screened off from between-cluster variation [56, 57]. The coefficient of the cluster-mean-centered variable for recent violence, for example, now represents the within-school association between suicidal ideation and recent violence. The five-category moderating variables, on the other hand, were entered into the regressions as raw dummies together with the school mean. Model (2), which was expressed by Eq 2 in the previous section, is now modified as:

$$
\begin{aligned}
\text{logit}(P(Y=1)) \\
&= (\beta_1 + \beta_7 STRESSveryhigh + \beta_8 STRESShigh + \beta_9 STRESSlow + \beta_{10} STRESSnone \\
&\quad + \beta_{11} STRESSsch.mean + u_{1j}) * VIOLENCE_{within} + \beta_2 STRESSveryhigh \\
&\quad + \beta_3 STRESShigh + \beta_4 STRESSlow + \beta_5 STRESSnone + \beta_6 STRESSsch.mean \\
&\quad + \text{other covariates} + u_{0j}
\end{aligned}
\tag{3}
$$

The "slope" S of within-cluster centered violence, i.e. the part within parentheses, is now a function of not just individual-level stress but also school-mean stress (plus the school-specific slope random effect $u_{1j}$). $\beta_7$ through $\beta_{10}$ represent the moderating effect of each stress category (vis-à-vis the reference category) on the within-school association between exposure to violence and suicidal ideation. $\beta_{11}$–the interaction coefficient between school mean stress and within-cluster centered violence–represents the moderating effect of school mean stress above

and beyond the moderating effect of individual-level stress [58], analogous to what is often called "contextual effect" in multilevel modeling, except that it is now applied to a moderation context. The school mean was computed by treating stress as a continuous variable (and likewise for economic status).

This modeling strategy marks a contrast with many previous works that employed multilevel regression to analyze the effect of "contextual" variables (such as group-mean stress) on suicidal vulnerability. Virtually all previous research of this kind examines the moderating effect of a contextual or climate variable by allowing the slope of the proximal stressor to vary as a function of the group-mean of the moderator variable but *not* the disaggregated individual-level moderator variable, that is, by having a cross-level interaction between the aggregated moderator variable and the individual-level proximal stressor without a corresponding individual-level interaction term [20, 24, 27, 59]. In Eq 3, this would correspond to omitting the four individual-level dummies for stress from the interaction with violence, i.e. fixing $\beta_7 - \beta_{10}$ to zero, and making inferences about the school-level contextual effect of stress on suicidal vulnerability based on $\beta_{11}$. Such a modeling strategy is completely unable to discriminate between the scenario in which the association between suicidal ideation and the proximal stressor varies in response to individual-level stress while remaining indifferent to the climate measured by group-mean stress and an alternative scenario in which it varies in response to such a climate while remaining indifferent to individual-level stress. Eq 3 is designed to capture the moderating effect of group-mean stress while accounting for the moderating effect of individual-level stress: If $\beta_{11}$ in Eq 3 is nonzero, it would indicate that two adolescents who have the same individual-level perceived stress but who differ in the average stress of their schoolmates would systematically differ in their strength of the association (as operationalized by the "slope" in Eq 3) between the recent occurrence of the proximal stressor and suicidal ideation.

The group means (required for group-mean centering) of the dummy variables for each type of crisis were estimated under partial pooling rather than computed as an arithmetic average under no pooling to avoid bias [60]. One way to estimate them under partial pooling is to run an intercept-only two-level logit regression and take the group-level residuals $u_j$ [20, 61, 62].

For individual $i$ in school $j$,

$$\text{Crisis}_{ij} \sim \text{Bernoulli}(p_{ij}) \tag{4}$$

$$\text{logit}(p_{ij}) = \alpha_j = \bar{\alpha} + u_j$$

$$u_j \sim \text{Normal}(0, \sigma)$$

Likewise, the cluster means of stress and economic status were estimated by coding them as continuous variables and running a two-level intercept-only mixed-effects regression similar to Eq 4 but with an identity link instead of logit. The cluster means of stress and economic status were standardized to have a standard deviation of 1.

Each crisis variable was allowed a random slope, which is represented by the random effect $u_{1j}$ in Eq 3. Estimation was implemented with the "brms" package, a convenient R interface to Stan [63, 64]. Convergence was not an issue, with all R-hat values well under 1.05. Weakly regularizing priors were used for all regression parameters.

Fig 5 shows the predicted probabilities for each crisis-stress and crisis-economic status interaction. There is little difference between Figs 2 and 5, indicating that the unexpected interaction effects seen in the national population also hold within local groupings. Indeed, the regression coefficient of the interaction term involving school mean stress or school mean

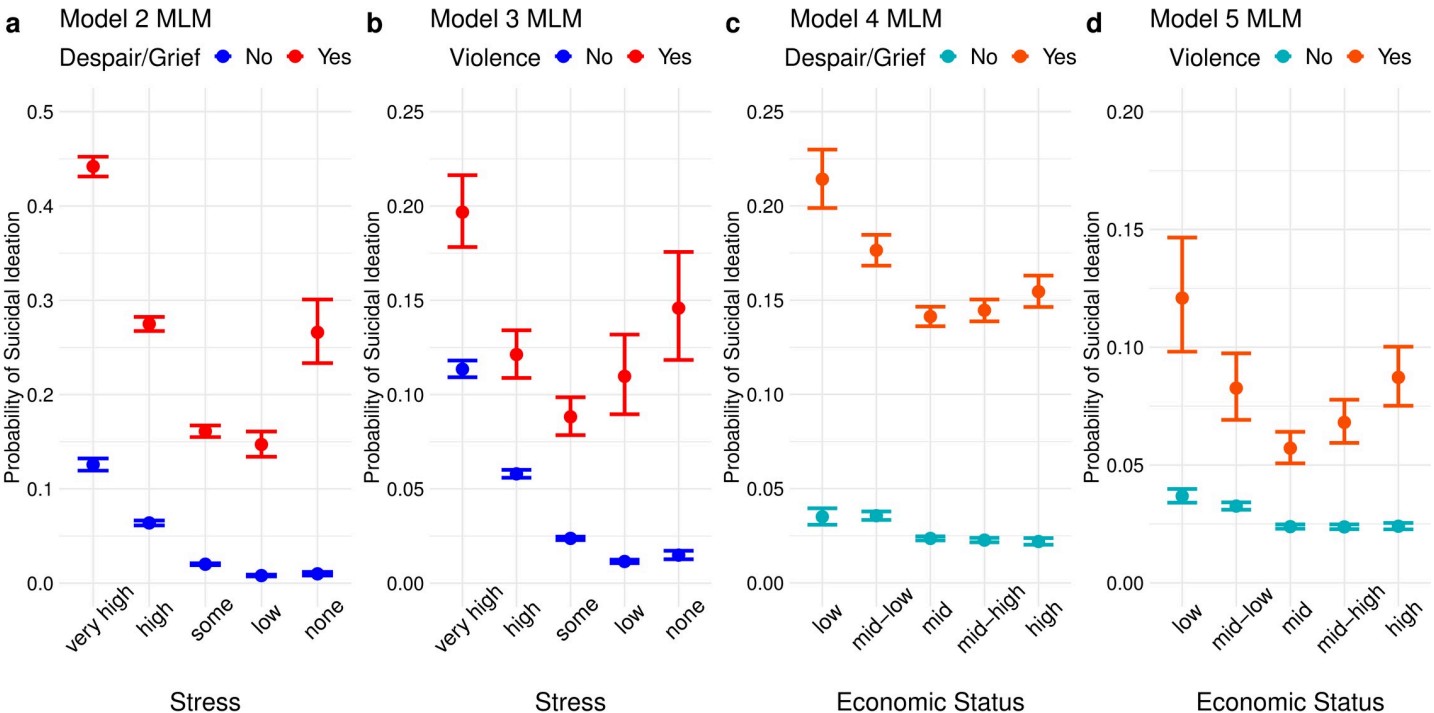

**Fig 5. Predicted probability by each category of stress and economic status, multilevel models.** All covariates not involved in the interaction terms were set to zero or the reference category. Random effects were set to zero. Error bars indicate 90% credible intervals.

economic status (analogous to $\beta_{11}$ in Eq 3), as well as the corresponding main effects (analogous to $\beta_6$ in Eq 3), were close to zero in all four models. The full results of the multilevel models are shown in S1 Code.

Overall, the above population-averaged and multilevel analysis suggests that stress, and to a lesser but still considerable extent economic status, statistically interact with psychosocial crisis and moderate its association to suicidal ideation. Contra Durkheim, their moderating effects seem to be operating mainly in virtue of individual-level perceived stress and economic status, and school-average perceived stress or economic status provides relatively little information on an adolescent's vulnerability to suicidal ideation once the corresponding individual-level variable is taken into account.

## Discussion

Previous research on suicidal vulnerability largely worked under the assumption that adolescents living in adverse psychosocial circumstances are generally more vulnerable to the suicide-inducing effects of a crisis. Within suicidology, this paper presents a first attempt to pose a theoretical and empirical challenge to such an assumption. Starting from the theoretical intuition that such a relationship may not always hold, using a large nationally representative sample of South Korean adolescents, I ran a series of logit regressions with a special focus on exploring non-linear interactions between a proximal stressor and self-rated stress or self-rated family economic status for predicting suicidal ideation. The most basic model without any interaction terms reconfirmed the conventional knowledge that higher stress and lower family economic status increase the risk of adolescent suicidal ideation. Yet, an exploration of complex and nonlinear interaction effects showed that *lower* usual stress *strengthens* the association between exposure to severe despair or violence in the past 12 months and suicidal

ideation when the association is conceptualized in terms of relative risk (Fig 3). A largely similar pattern was observed with self-rated family economic status in lieu of stress except for the sudden reversal in the lowest category. Speaking in terms of absolute probability, for both stress and economic status, the predicted probability of suicidal ideation made a clear U- or J-shape given the occurrence of either type of crisis (Figs 2 and 5). This is in stark contrast with the predicted probabilities in the absence of a crisis that (nearly) monotonically decreases with improving levels of stress or economic status. The difference in the predicted probabilities is the smallest somewhere around the middle category of stress and economic status and gets wider closer to the extremes, indicating a U-shaped additive interaction effect (Fig 4). In sum, both in terms of relative risk and risk difference, it is not necessarily the case (or, it is often not the case) that being in a higher stress category or having a lower perceived family economic status strengthens the association between recent crisis and suicidal ideation.

These results may potentially be explained by the burgeoning psychological literature on stress inoculation: Having been exposed to considerable psychological or ecological life adversity results in mental resilience in the face of adverse stimuli, except at sufficiently high levels of exposure that tend to result in vulnerability rather than resilience. The general upward trend in relative risk by improving levels of stress or economic status (Fig 3) is consistent with the theory that having lived a life with low levels of adversity and few opportunities for steeling leads to vulnerability for adolescents. The sudden reversal at the lowest category of economic status is indicative of mental attrition for adolescents who are continuously exposed to the worst socioeconomic circumstances that do not allow for steeling, a finding that is especially salient considering the high base rate of suicidal ideation in this category. The pattern in risk difference (Fig 4) is also consistent with the insight that moderate levels of life adversity lead to steeling, while extremities in either direction result in weakness.

These findings unmistakably go against the prevalent understanding of suicidal vulnerability, but its discrepancy with the existing empirical literature should not be overstated. As for stress, there is a paucity of empirical research that directly examines the relationship between stress and vulnerability to suicide-related outcomes in the face of a crisis, and the widespread understanding of a monotonically positive association appears to be an assumption rather than knowledge with a thorough empirical justification. There is a slightly more sizable accumulation of research on how one's vulnerability to suicidal outcomes (or depression) in the face of proximal stressors are moderated by one's (socio)economic status. However, these works either focus on poverty [24, 27] or study the effect of economic status across its entire range but model interaction effects linearly [26]. The finding that a low (when measured in relative risk/odds) or low and medium-low (when measured in risk difference) level of self-rated family economic status strengthens the association between recent crisis and suicidal ideation is consistent with previous research that found that poverty is positively associated with suicidal vulnerability.

Finally, a mention should be made about the KYRBS dataset used for this study. It has a clear advantage over most other public health surveys in terms of size, response rate, and missing values. The large size enabled fitting a complicated regression model without having to assume any particular shape of the main and interaction effects *ex-ante*. However, it is also the source of many of the limitations of this paper. The most conspicuous problem is that it lacks the precise and/or time-varying measurements compared to many other public health datasets and offers relatively rough measurements at the individual level. Notably, some variables such as family suicide and previous suicide attempt were unavailable. Although the stability of the U-shaped pattern through a large variety of controls suggests this is a robust relationship, the lack of certain controls leaves room for additional verification. As a related issue, the proposed causal mechanisms responsible for the rather surprising findings are speculative and will

require a dataset with more purposefully designed variables for verification. All these limitations call for further research using different datasets collected from other populations. Still, the clarity and consistency of the patterns in the absolute and relative probabilities of suicidal ideation for each category of perceived stress and economic status reported above strongly suggests that the relationship between psychological or economic adversity and vulnerability to suicidal outcomes is more complex than is often assumed in the existing literature.

## Supporting information

**S1 Fig. Logit regression coefficients of interaction terms.**
(TIF)

**S1 Code. Annotated R script presentation in.html format for replication.**
(HTML)

## Acknowledgments

I would like to thank Prof. Matthew Lange, Prof. Peter McMahan, Prof. Eran Shor, and Prof. Axel van den Berg (in alphabetical order) for their helpful comments on earlier drafts of this paper.

## Author Contributions

**Conceptualization:** Tay Jeong.

**Data curation:** Tay Jeong.

**Formal analysis:** Tay Jeong.

**Investigation:** Tay Jeong.

**Methodology:** Tay Jeong.

**Project administration:** Tay Jeong.

**Resources:** Tay Jeong.

**Software:** Tay Jeong.

**Supervision:** Tay Jeong.

**Validation:** Tay Jeong.

**Visualization:** Tay Jeong.

**Writing – original draft:** Tay Jeong.

**Writing – review & editing:** Tay Jeong.

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
