## [Decision Letter · Decision Letter 0]

15 Feb 2021

PONE-D-20-36724

Do More Stress and Lower Family Economic Status Increase Vulnerability to Suicidal Ideation? Evidence of a U-shaped relationship in a Large Cross-sectional Sample of South Korean Adolescents

PLOS ONE

Dear Dr. Jeong,

Thank you for submitting your manuscript to PLOS ONE. After careful consideration, we feel that it has merit but does not fully meet PLOS ONE’s publication criteria as it currently stands. Therefore, we invite you to submit a revised version of the manuscript that addresses the points raised during the review process.

Please be aware that submitting a revision does not guarantee acceptance.

We look forward to receiving your revised manuscript.

Kind regards,

Vincenzo De Luca

Academic Editor

PLOS ONE

Journal Requirements:

3. We note you have included a table to which you do not refer in the text of your manuscript. Please ensure that you refer to Table 1 in your text; if accepted, production will need this reference to link the reader to the Table.

Reviewers' comments:

Reviewer's Responses to Questions

**Comments to the Author**

1. Is the manuscript technically sound, and do the data support the conclusions?

Reviewer #1: Yes

Reviewer #2: Yes

2. Has the statistical analysis been performed appropriately and rigorously? 

Reviewer #1: Yes

Reviewer #2: Yes

3. Have the authors made all data underlying the findings in their manuscript fully available?

Reviewer #1: Yes

Reviewer #2: Yes

4. Is the manuscript presented in an intelligible fashion and written in standard English?

Reviewer #1: Yes

Reviewer #2: No

5. Review Comments to the Author

Reviewer #1: This is a great study examining the relationship between stress, SES, and suicidal ideations in a sample of Korean adolescents. A few minor revisions recommendations. Please clarify what "Grades rather than breaks" mean in more detail in Line 56 of your introduction. It is a bit jargony at times and difficulty for the lay-audience to understand. Please also speak a little more to the stress inoculation literature and hypothesis in your discussion. Please discuss how your current findings fit in with the existing literature, and any differences/additions that this study contributes. In your data analysis, I would recommend including sample descriptive statistics in relation to demographic data (i.e., specific SES/income distributions; age range; number of females to males; etc). Did you control for family history of suicides given that family history and prior suicide attempts are strong predictors of current suicidal ideations and plans? Please describe more how you have statistically controlled for or taken these variables into account. Overall, great findings. It was a pleasure to review.

Reviewer #2: This manuscript reports on an interesting and innovative analysis. The findings suggest that associations between environmental stressors (i.e., stress/SES), the experience of social loss or violence, and suicidal ideation are complex. Intriguingly, in the presence of social loss/violence, the risk of suicidal ideation may be lowest when there is some stress or at mid-levels of SES (and not when there is no perceived stress or at highest SES levels, which is what one might expect).

This study has strengths, such as its large sample size, which allows for investigation into nonlinear or interaction effects. Exploring these effects is not common and very much lacking in this type of work. I hope this paper, if published, would encourage other researchers to replicate and extend on the author’s finding. However, my main concern with this manuscript is the clarity of the writing and description, which I believe needs to be addressed prior to publication.

1. Is the manuscript technically sound, and do the data support the conclusions?

The analytical strategy appears to be sound (and the author takes care to explain and justify analytical decisions). In some places, however, the technical soundness is somewhat difficult to assess, which may be due to the lack of clarity in the writing (more on this in #4 below).

For example, I generally understand that lines 294-320 describe the novel finding that being in a higher economic category or having less stress increases the likelihood of suicidal ideation when paired with social loss or violence in the past year. However, I found the description (and Figure 2) hard to follow (and in fact, found consulting S1 to be useful, though S1 has a typo, where Despair x Econ Status “mid-high” appears twice, and should be “high” in one instance).

I find Figure 3 most useful and interesting (more than Figure 2, which may fit better in the Supplement). Figure 3 still demonstrates those with low SES/very high stress, paired with an experience of social loss or violence, are at highest likelihood of suicidal ideation, which can be acknowledged a bit more (or more clearly). But the U or J-shaped pattern at remaining levels of SES/stress (paired with despair/violence) is fascinating and depicted well with this Figure.

For Figure 5, it can be described more explicitly that a smaller distance/difference indicates lower risk of suicidal ideation (with middle categories of stress or SES at lowest risk).

Little descriptive information about the sample is provided. It would be useful to present this information in a table along with the covariate information (e.g., to indicate M (SD) or N’s and %’s for age, sex, grade, residential type, etc.) In fact, information from Figure 1 might also be better presented as part of this table, rather than separate graphs for each variable of interest.

The explanation for how and why the multilevel models are conducted is a bit hard to follow – what are “like associations”? How does this analysis assess whether stress has contextual effects on suicidal ideation? Also, I understand that the author quantified these variables at school-level (e.g., mean stress), and included this as an interaction with social loss or violence in the models. I also see that this does not change the main findings, but perhaps some of the technical details about this in the results can be moved to the statistical analysis section to make these results easier to read and follow.

The analysis was not replicated in a different dataset and given that this paper reports on a secondary analysis, perhaps this is not a barrier to publication. However, the author might note in the discussion that the findings require replication.

A major strength of this paper is the demonstration that considering interaction and nonlinear effects is valuable and provides insight into more complex relationships between adversity and suicidal ideation (since the main effects reported are what one would expect, but when interactions are considered, there are some surprising and interesting effects). I think this could be noted or highlighted more in the discussion.

Finally, some of the language in the discussion may be a bit stigmatizing (e.g., the term “mental weakening” to imply suicidal ideation).

*2. Has the statistical analysis been performed appropriately and rigorously?

Yes, this appears to be the case. The author conducted diagnostics to ensure the results from the regression models were sound (and that findings were not different in multilevel models is useful to know).

*3. Have the authors made all data underlying the findings in their manuscript fully available?

Yes, the R code is provided, as well as a description of how to access the raw data.

*4. Is the manuscript presented in an intelligible fashion and written in standard English?

In my view, this is the weakest part of the manuscript, as the writing and descriptions of the methods, statistical analysis, and results is unclear.

The author should not interpret the variables and just report them as they are (e.g., instead of saying “more favourable levels of perceived stress” or “better stress category” or “improving levels of family SES”, say “lower levels of perceived stress” or “higher levels of family SES”)

The author refers to suicidal ideation or severe grief/despair as being “recent” whereas the time range should just be reported as is (i.e., “over the past year/12 months”)

The author refers to the variables inconsistently (“psychosocial crisis/grief/despair/violence” or “suicidal ideation/vulnerability”). I would recommend selecting one term for each variable and using it consistently throughout.

In line 222, “well-being” is introduced, but it hasn’t been explained or defined (unless this refers to the “stress” variable? Or “self rated health”?)

Some of the statements are unclear, e.g., in the abstract, the author writers “a supplementary exercise suggests that the identified moderation effects operate mainly in virtue of individual level traits in the relative absence of contextual influences at the school level”, which is hard to follow or understand what was done.

The paragraph on strain theory (lines 136-145) assumes that readers are familiar with classical strain theory, whereas this theory should be defined and explained more. It is also a bit unclear why this distinction between expectation and reality as a strain is mentioned here. Please make a clearer link to stress inoculation earlier on.

Lines 156-160 present concepts/ideas that are not too clearly explained (does the author just mean to say that people who are financially well off have more to lose in a financial crisis?)

The word choice used in some cases is a bit strange, e.g., line 74 (“a smaller group of research”), line 78 (“more severely”) …etc.

In general, the writing could be more concise and could use editing. There are some grammatical issues (with plurals, e.g., in the abstract “using a large survey data”, an “and” missing on line 65, lines 90-91, line 117 and line 485 seem to have errors or typos, some sentences are long and hard to read/follow, e.g., lines 136-140)

Line 126 – PTS is introduced but not explained (do you mean post traumatic stress?)

Much of the information about statistical analyses is currently described in the “Results” and may fit better in the “Statistical analysis” section.

6. PLOS authors have the option to publish the peer review history of their article (what does this mean?). If published, this will include your full peer review and any attached files.

Reviewer #1: No

Reviewer #2: No

---

## [Author Response · Author response to Decision Letter 0]

25 Mar 2021

PONE-D-20-36724

March 24, 2021

Dear Dr. Vencenzo De Luca,

Editor of PLOS ONE,

Thank you for allowing me to revise and improve my manuscript. I was lucky to receive helpful comments from the two anonymous reviewers. This letter describes the changes I have made to the manuscript in response to the reviewers’ comments. It also explains why I did not make the proposed revisions regarding a couple of issues.

This letter is organized in two parts: The first part broadly describes the content of the revision without citing each point made by the two reviewers. The second part quotes reviewer comments one by one and presents my response.

Part 1

Most of the criticisms I received from the reviewers were related to the clarity/efficiency of writing and presentation. To list some of the notable changes I made, I

1. Deleted several paragraphs near the beginning of the section “Population-averaged models” (lines 319-346 in the markup draft), which had originally explained how to interpret regression coefficients of the interaction terms. The discussion of the models with interaction effects now leads directly to the predicted probability plot (Fig 2), which one reviewer said was the most helpful and interesting;

2. Moved the regression coefficient plot in this part (previously Fig 2) to the supplementary material section following the suggestion of the second reviewer;

3. Added a paragraph briefly summarizing the stress inoculation theory and General Strain Theory for additional clarity (lines 169-177);

4. Brought the regression table (previously S1 Table) into the main text and made it Table 1 following Requirement 3 in your decision email and the second reviewer’s comments;

5. The first reviewer asked me if I controlled for family suicide and prior suicide attempt in my study and suggested that I describe this clearly in the text. This information was unfortunately not available in this dataset, so it was not possible to add them as additional controls. In response, I acknowledged this as a limitation of the study in the “Discussion” section and related it to the need for further research and verification.

6. Made a number of other minor modifications throughout the text according to the very detailed recommendations of the second reviewer and PLOS ONE format requirements. In particular, I standardized/unified terminologies and deleted confusing terms (e.g. well-being).

There were also a couple of instances in which I, for the time being, did not follow the recommendation of at least one reviewer. 

First, instead of adding a separate table showing the descriptive statistics for all variables used, I added this information directly to the relevant parts of the main text of the “Variables” section. All key variables used in this study were categorical variables, and I think their distributions are more effectively communicated through histograms rather than tables. The distribution of the remaining (control) variables could easily be communicated in the main text without having to add a separate table. 

Second, I kept the statistical/technical discussion in the section “Multilevel models” roughly the same length despite one reviewer’s suggestion to either shorten it and/or move it to a new section. In my humble opinion, laying out the technical details constitute an integral part of the presentation here, especially considering the complexity of the models used and the sensitivity of the results/interpretations to model specification. Instead of cutting or relocating, I modified the corresponding text to increase clarity and readability. 

Part II

Reviewer 1

Please clarify what "Grades rather than breaks" mean in more detail in Line 56 of your introduction.

- Added a clarifying phrase in parentheses

Please also speak a little more to the stress inoculation literature and hypothesis in your discussion.

- Added a paragraph summarizing the main idea behind the stress inoculation hypothesis and General Strain Theory. I could have given more references and go into more detail for each theory but decided not to for the time being as I feared it would distract the reader.

Please discuss how your current findings fit in with the existing literature, and any differences/additions that this study contributes.

- I initially thought this was already quite clear, but the reviewer’s comments made me realize it may not have been spelled out in enough clarity and visibility. I modified the discussion in a way that more explicitly describes the contribution of this article. 

Did you control for family history of suicides given that family history and prior suicide attempts are strong predictors of current suicidal ideations and plans? Please describe more how you have statistically controlled for or taken these variables into account.

- I could not address fully address this issue as the limitations were “baked in” the dataset itself. I added a couple of sentences in the discussion (lines 548-554) to acknowledge this limitation more clearly and suggest that the findings of this research, while robust within the available range of controls, could benefit from further verification and research.

Reviewer 2

For example, I generally understand that lines 294-320 describe the novel finding that being in a higher economic category or having less stress increases the likelihood of suicidal ideation when paired with social loss or violence in the past year. However, I found the description (and Figure 2) hard to follow (and in fact, found consulting S1 to be useful, though S1 has a typo, where Despair x Econ Status “mid-high” appears twice, and should be “high” in one instance).

- I removed the part that includes lines 294-320 for simplicity. I relocated Fig 2 to the supplementary materials section and brought S1 Table into the main text as Table 1. I corrected the typo.

Figure 3 still demonstrates those with low SES/very high stress, paired with an experience of social loss or violence, are at highest likelihood of suicidal ideation, which can be acknowledged a bit more (or more clearly).

- In the revised draft I acknowledged this explicitly in the text (lines 356-358 of the manuscript with track changes).

For Figure 5, it can be described more explicitly that a smaller distance/difference indicates lower risk of suicidal ideation (with middle categories of stress or SES at lowest risk).

- I added several lines for additional clarification to the corresponding part.

Little descriptive information about the sample is provided. It would be useful to present this information in a table along with the covariate information (e.g., to indicate M (SD) or N’s and %’s for age, sex, grade, residential type, etc.) In fact, information from Figure 1 might also be better presented as part of this table, rather than separate graphs for each variable of interest.

- As mentioned in Part 1 of this letter, I addressed the issue by adding the relevant information directly into the main text (lines 384-393 in the revised manuscript with tracket changes).

The explanation for how and why the multilevel models are conducted is a bit hard to follow – what are ‘like associations’? How does this analysis assess whether stress has contextual effects on suicidal ideation? Also, I understand that the author quantified these variables at school-level (e.g., mean stress), and included this as an interaction with social loss or violence in the models. I also see that this does not change the main findings, but perhaps some of the technical details about this in the results can be moved to the statistical analysis section to make these results easier to read and follow.

- I modified ambiguous word choices such as “like associations.” I tried to add clarity to the model by using a more intuitively clear Equation (Equation 3). I also added several lines throughout this part to explain and justify my model specification. I found it somewhat hard to specify the model and interpret the results without a moderate degree of technical discussion and chose not to relocate it to a separate section for statistical methods for the time being. Lines 459-471 (in the manuscript with track changes) could probably be safely moved to a different section without compromising the flow in this section, but I thought it is a bit too short to be included as a separate section

However, the author might note in the discussion that the findings require replication.

- Added a line in the last paragraph of the “Discussion” section.

A major strength of this paper is the demonstration that considering interaction and nonlinear effects is valuable and provides insight into more complex relationships between adversity and suicidal ideation (since the main effects reported are what one would expect, but when interactions are considered, there are some surprising and interesting effects). I think this could be noted or highlighted more in the discussion.

- Added several lines in the first paragraph of the “Discussion” section to more clearly summarize the unique contribution of this article.

Finally, some of the language in the discussion may be a bit stigmatizing (e.g., the term “mental weakening” to imply suicidal ideation).

- I changed “weakening” to “attrition.” Although synonymous, I thought avoiding the word “weak” might still make it sound less stigmatizing.

The author should not interpret the variables and just report them as they are (e.g., instead of saying “more favourable levels of perceived stress” or “better stress category” or “improving levels of family SES”, say “lower levels of perceived stress” or “higher levels of family SES”)

- I got rid of confusing expressions like “favorable stress” or “better economic status.”

The author refers to suicidal ideation or severe grief/despair as being “recent” whereas the time range should just be reported as is (i.e., “over the past year/12 months”)

- I switched “recent” to “in the past 12 months” in multiple locations. I kept recent in a few places where adding “in the past 12 months” is a bit too wordy in the context. In my humble opinion, I think it would not be too problematic to use it when it is clear from the surrounding context that it refers to the past 12 months, especially as it has been defined as such in the “Variables” section.

The author refers to the variables inconsistently (“psychosocial crisis/grief/despair/violence” or “suicidal ideation/vulnerability”). I would recommend selecting one term for each variable and using it consistently throughout.

- Psychosocial crisis is a broader event category to which severe despair and experience of violence belong, so I would say that they are not the same. Suicidal ideation and vulnerability are also different concepts, as the latter captures how much the effect of a proximal stressor on suicidal ideation is amplified/dampened by a background factor. Regarding the latter point, I added a line in the first paragraph of the introduction to clarity that vulnerability is to be understood as an interaction effect. 

In line 222, “well-being” is introduced, but it hasn’t been explained or defined (unless this refers to the “stress” variable? Or “self rated health”?)

- I deleted the word well-being.

Some of the statements are unclear, e.g., in the abstract, the author writers “a supplementary exercise suggests that the identified moderation effects operate mainly in virtue of individual-level traits in the relative absence of contextual influences at the school level”, which is hard to follow or understand what was done.

- I thought the lack of clarity comes from the phrase “individual-level traits.” (What individual-level traits? Hard to know before reading the main text). I changed it to “individual-level stress or family economic status.”

The paragraph on strain theory (lines 136-145) assumes that readers are familiar with classical strain theory, whereas this theory should be defined and explained more. It is also a bit unclear why this distinction between expectation and reality as a strain is mentioned here. Please make a clearer link to stress inoculation earlier on. Lines 156-160 present concepts/ideas that are not too clearly explained (does the author just mean to say that people who are financially well off have more to lose in a financial crisis?)

- Classical strain theory does not have a direct application to the case under consideration, so I fear that describing it would only distract the reader from the main flow of the article. I think the reviewer made this criticism because I mentioned classical strain theory without any introduction at all. In retrospect, it was not necessary to introduce classical strain theory in my discussion of general strain theory. I therefore simply deleted any mention of classical strain theory. 

- I also added a paragraph briefly summarizing the main idea of the stress inoculation hypothesis and general strain theory. I think this may also add a bit more clarity to lines the part that the reviewer mentioned (which is line 166-169 in the revised manuscript with track changes).

The word choice used in some cases is a bit strange, e.g., line 74 (“a smaller group of research”), line 78 (“more severely”) …etc. In general, the writing could be more concise and could use editing. There are some grammatical issues (with plurals, e.g., in the abstract “using a large survey data”, an “and” missing on line 65, lines 90-91, line 117 and line 485 seem to have errors or typos, some sentences are long and hard to read/follow, e.g., lines 136-140)

- I modified the words/sentences appropriately and ran another round of grammar check, both manual and automatic. 

Line 126 – PTS is introduced but not explained (do you mean post traumatic stress?)

- I changed it to “post-traumatic stress symptoms.”

Much of the information about statistical analyses is currently described in the “Results” and may fit better in the “Statistical analysis” section.

- This relates to some of the points I made earlier in this letter. Currently, there is no section that specifically lays out and explains technical details of the statistical methods used. After deleting a large portion in the section “Population-averaged models” (lines 319-346 in the manuscript with track changes), I thought there is not much content left that could be relocated to an entirely new section. This is something I could change if necessary, but my current judgment is that it might be better not to add a separate section that only contains technical details.

I would like to thank Dr. De Luca again for considering my article for publication in PLOS ONE and the two reviewers for their helpful comments.

Yours sincerely,

Tay Jeong

---

## [Decision Letter · Decision Letter 1]

6 Apr 2021

PONE-D-20-36724R1

Do More Stress and Lower Family Economic Status Increase Vulnerability to Suicidal Ideation? Evidence of a U-shaped relationship in a Large Cross-sectional Sample of South Korean Adolescents

PLOS ONE

Dear Dr. Jeong,

Thank you for submitting your manuscript to PLOS ONE. After careful consideration, we feel that it has merit but does not fully meet PLOS ONE’s publication criteria as it currently stands. Therefore, we invite you to submit a revised version of the manuscript that addresses the points raised during the review process.

We look forward to receiving your revised manuscript.

Kind regards,

Vincenzo De Luca

Academic Editor

PLOS ONE

Journal Requirements:

Reviewers' comments:

Reviewer's Responses to Questions

**Comments to the Author**

1. If the authors have adequately addressed your comments raised in a previous round of review and you feel that this manuscript is now acceptable for publication, you may indicate that here to bypass the “Comments to the Author” section, enter your conflict of interest statement in the “Confidential to Editor” section, and submit your "Accept" recommendation.

Reviewer #1: (No Response)

2. Is the manuscript technically sound, and do the data support the conclusions?

Reviewer #1: Yes

3. Has the statistical analysis been performed appropriately and rigorously? 

Reviewer #1: Yes

4. Have the authors made all data underlying the findings in their manuscript fully available?

Reviewer #1: Yes

5. Is the manuscript presented in an intelligible fashion and written in standard English?

Reviewer #1: Yes

6. Review Comments to the Author

Reviewer #1: Thank you for submitting the revised manuscript and incorporating many of the comments/edits that the reviewers have recommended. It is a significantly improved draft. I just have 2 minor edits for you to address:

1) Line 206: Please describe how you handled the missing values for those variables that had them? What were the statistical decisions made for those variables with missing values?

2) Lines 476/480: See if you can rephrase the "harder" and "softer" referenced in this paragraph of the discussion section. It is too colloquial and contains certain judgements/connotations that did not fit the rigour of a scientific discussion. I understood what you meant by it, but see if you can reword it and explain it in a different way for the lay audience.

7. PLOS authors have the option to publish the peer review history of their article (what does this mean?). If published, this will include your full peer review and any attached files.

Reviewer #1: No

---

## [Author Response · Author response to Decision Letter 1]

7 Apr 2021

Response to Reviewers

PONE-D-20-36724R1

April 6, 2021

Dear Dr. Vincenzo De Luca,

Editor of PLOS ONE,

I have received minor but apt criticisms from the anonymous reviewer and made certain changes to the manuscript. Listed below are direct citations of the reviewer’s comments followed by my responses.

1) Line 206: Please describe how you handled the missing values for those variables that had them? What were the statistical decisions made for those variables with missing values?

Author’s Response: Thank you for pointing this out. In response to this comment, I explained that there are almost no missing values in the entire dataset because students were required to answer all questions in the online survey, and that the rare cases of missing responses come from a handful of confidential questions in the 2019 survey. I also clearly declared that missing data posed no concerns for variable choice given the excellent availability of virtually all relevant variables. 

2) Lines 476/480: See if you can rephrase the "harder" and "softer" referenced in this paragraph of the discussion section. It is too colloquial and contains certain judgements/connotations that did not fit the rigour of a scientific discussion. I understood what you meant by it, but see if you can reword it and explain it in a different way for the lay audience.

Author’s Response: I deleted these words and used expressions that I believe are more suitable for academic communication.

I would like to thank Dr. De Luca and the anonymous reviewer again for helping me improve and (potentially) publish my paper at PLOS ONE.

Yours sincerely,

Tay Jeong

---

## [Decision Letter · Decision Letter 2]

14 Apr 2021

Do More Stress and Lower Family Economic Status Increase Vulnerability to Suicidal Ideation? Evidence of a U-shaped relationship in a Large Cross-sectional Sample of South Korean Adolescents

PONE-D-20-36724R2

Dear Dr. Jeong,

We’re pleased to inform you that your manuscript has been judged scientifically suitable for publication and will be formally accepted for publication once it meets all outstanding technical requirements.

Kind regards,

Vincenzo De Luca

Academic Editor

PLOS ONE

Additional Editor Comments (optional):

Reviewers' comments:

Reviewer's Responses to Questions

**Comments to the Author**

1. If the authors have adequately addressed your comments raised in a previous round of review and you feel that this manuscript is now acceptable for publication, you may indicate that here to bypass the “Comments to the Author” section, enter your conflict of interest statement in the “Confidential to Editor” section, and submit your "Accept" recommendation.

Reviewer #1: All comments have been addressed

2. Is the manuscript technically sound, and do the data support the conclusions?

Reviewer #1: Yes

3. Has the statistical analysis been performed appropriately and rigorously? 

Reviewer #1: Yes

4. Have the authors made all data underlying the findings in their manuscript fully available?

Reviewer #1: Yes

5. Is the manuscript presented in an intelligible fashion and written in standard English?

Reviewer #1: Yes

6. Review Comments to the Author

Reviewer #1: (No Response)

7. PLOS authors have the option to publish the peer review history of their article (what does this mean?). If published, this will include your full peer review and any attached files.

Reviewer #1: No

---

## [Editor Report · Acceptance letter]

16 Apr 2021

PONE-D-20-36724R2 

Do more stress and lower family economic status increase vulnerability to suicidal ideation? Evidence of a U-shaped relationship in a large cross-sectional sample of South Korean adolescents 

Dear Dr. Jeong:

I'm pleased to inform you that your manuscript has been deemed suitable for publication in PLOS ONE. Congratulations! Your manuscript is now with our production department. 

Kind regards, 

on behalf of

Dr. Vincenzo De Luca 

Academic Editor

PLOS ONE